# Overimitation in Dogs: Is There a Link to the Quality of the Relationship with the Caregiver?

**DOI:** 10.3390/ani12030326

**Published:** 2022-01-29

**Authors:** Ludwig Huber, Denise Kubala, Giulia Cimarelli

**Affiliations:** 1Clever Dog Lab, Comparative Cognition, Messerli Research Institute, University of Veterinary Medicine Vienna, Medical University of Vienna, University of Vienna, 1210 Vienna, Austria; denise.kubala@vetmeduni.ac.at (D.K.); giulia.cimarelli@vetmeduni.ac.at (G.C.); 2Domestication Lab, Konrad Lorenz Institute of Ethology, University of Veterinary Medicine Vienna, 1210 Vienna, Austria

**Keywords:** overimitation, dogs, affiliation, relationship, dog–human interaction

## Abstract

**Simple Summary:**

Humans, but no other primates, show overimitation: the copying of causally irrelevant or non-functional actions. Recent studies have provided evidence for overimitation in canines, but mainly when their human caregiver—not an unfamiliar person—is demonstrating the irrelevant action. Therefore, we hypothesized that dogs show overimitation as a result of affiliation. Here, we tested if the eagerness to overimitate is influenced by the relationship quality between dog and caregiver by measuring, on the one hand, their relationship and on the other hand, the overimitation tendency. Although our overall results were not significant, our data might suggest that, on average, the dogs who overimitated might also show more referential and affiliative behaviours towards the owner (like gazing, synchronization and greeting) than dogs who showed less or no copying of the irrelevant action. Possible reasons for these negative findings are discussed, together with the implications for dog owners.

**Abstract:**

Overimitation, the copying of causally irrelevant or non-functional actions, is well-known from humans but completely absent in other primates. Recent studies from our lab have provided evidence for overimitation in canines. Previously, we found that half of tested pet dogs copied their human caregiver’s irrelevant action, while only few did so when the action was demonstrated by an unfamiliar experimenter. Therefore, we hypothesized that dogs show overimitation as a result of socio-motivational grounds. To test this more specifically, here we investigated how the relationship with the caregiver influenced the eagerness to overimitate. Given the high variability in the tendency to overimitate their caregiver, we hypothesized that not only familiarity but also relationship quality influences whether dogs faithfully copy their caregiver. For this purpose, on the one hand we measured the overimitation tendency (with the same test as in the two studies before) and on the other hand the relationship quality between the dogs and their caregivers. Although we found no significant correlation between the two test results, our data might suggest that, on average, dogs who overimitated seemed to show more referential and affiliative behaviours towards the owner than dogs who showed less or no copying of the irrelevant action. Notably, as a group, those dogs that showed the highest level of copying accuracy of the irrelevant action showed the highest level of gazing and synchronization towards the owner.

## 1. Introduction

The origin and nature of the close relationship between human caregivers and their pet dog(s), being popularized as “man’s best friend”, have been a long-standing focus of scientific inquiry. It has been hypothesized that such cooperative relationships were made possible by the “exploitation” of a pre-existing tolerant and cooperative attitude in canines [1]. Additionally, during domestication, humans have selected dogs for compliant cooperation by favouring dogs who are less fearful and more submissive to minimize conflict over resources [2]. According to this hypothesis, dogs do cooperate with humans, but mostly following the human’s lead. In addition to these social tendencies, it has been argued that dogs have developed sensory and cognitive mechanisms that enhance cooperation, such as a sensitivity for specific aspects of human communication [3].

From an emotional point of view, the relationship between companion dogs and their human caregivers can take a form that resembles an infant attachment bond. Dogs are dependent on human care and their behaviour seems specifically geared to engage their human’s caregiving system [4,5,6,7,8]. As a consequence, dogs do not just seek and maintain contact with their human partner, their exploration and problem-solving abilities seem to be strongly facilitated by the intimate relationship [9,10] and, like children, they can interpret a test situation as being a social, communicative game [11]. This might also explain why dogs are able to synchronize their behaviour with that of their owners without any immediate reinforcement, or are able to anticipate their owner’s actions to some extent [12,13].

The close affiliative relationship between dogs and their caregiver is also likely to change the way they learn from them. Dogs learn from their caregiver in both active and passive manners. On the one hand, they are educated in direct, teaching-like manners of what to do and what not to do [14], which allows learning about rules, causally opaque actions and conventions. Especially if performed by the caregiver in an ostensive, teaching-like manner, dogs learn to obey and follow [15,16]. Nevertheless, there are also many situations in which the dog learns in a passive, latent manner. This form of demonstrator-matching behaviour is based on mechanisms like priming, social facilitation and enhancement. Furthermore, playing games with the dog may facilitate its learning, especially by an increase in ‘obedient attentiveness’ [17]. In combination, these factors have led us to assume that pet dogs living in an attachment-like relationship with their human caregiver tend to learn from them like children from their parents. Dogs may learn not by copying only functional and relevant actions in a goal-directed, efficiency-based manner but more in a play-like and normative manner by also copying inefficient, non-functional or causally opaque actions, i.e., overimitation [18].

In previous studies, dogs were able to overimitate when the owner [19], or a familiar experimenter with whom the animals had previous positive interactions with [20], was a demonstrator. However, dogs were much less likely to do so when the demonstrator was unfamiliar and had not previously interacted with them [21]. These contrasting results highlight the importance of demonstrator familiarity in overimitation tasks. Furthermore, the differences in the dogs’ performance between subjects in these studies suggest that, also, the individual relationship with the demonstrator might play a role. In fact, we observed more variation in the Huber et al. [19] study, in which dogs were tested with owners, than in the Johnston et al. [20] study, in which interactions with the experimenter were standardised, and thus the same for all dogs. This led us to hypothesize a positive correlation between the quality of the relationship between dogs and their caregivers, and the likeliness of the dogs to copy causally irrelevant, non-functional actions of their caregivers: the better the relationship with the human partner, the more likely the dog will overimitate. Although, at this point, we can only speculate about the underlying causality of this correlation, we assume that a better relationship would lead to more attention towards the caregiver’s actions, more willingness to comply and a stronger normative effect [18].

In order to test our hypothesis, we confronted a new sample of dogs with two tests; the overimitation test of Huber et al. [19,21] and the relationship test of Cimarelli et al. [22,23]. The latter measured dogs’ behaviour in a test battery that was developed to simulate a series of situations that pet dogs, during their everyday life, may naturally face with their owners. With this test, the authors could identify components of pet dogs’ relationships that are associated with the attachment system and characterize the different types of relationships dogs have established with their human partner. This was achieved by confronting dog–owner dyads with situations that involved a chance to explore a novel environment, a separation and reunion phase and the appearance of a novel, potentially fear-evoking object. Results revealed that pet dogs’ relationships are characterized by three components: (1) *reference* to the human partner, e.g., gaze alternation, greeting, (2) *affiliation*, e.g., affiliative or synchronized behaviours and (3) *stress*, i.e., stress-related behaviours and marking. Here, we analysed the same behaviours as a way to explore possible links to individual dog overimitation performance, i.e., the copying accuracy of the irrelevant actions.

In the overimitation test, which was conducted before the relationship test on the same day, we confronted the dogs first with the demonstration of a causally superfluous, functionally opaque or irrelevant action, followed by an effect-relevant or causally relevant action and, finally, the effect or goal. The order of these two actions followed the classical procedure (e.g., [20,24]). The relevant action is the one that allows the goal to be reached; it is logically necessary that the irrelevant action happens before that. Furthermore, we did not find significant differences if the order was reversed [19,21]. To increase the causal transparency of the ‘irrelevant’ action, it was spatially separated from the relevant action and consisted of the touching of colour dots on two sheets of paper. The relevant action consisted of pushing a sliding door to the side and, thereby, opening a food compartment. The main variable of this test was the copying accuracy of the irrelevant action, which varied in two levels (touching only one or both dots). Following our hypothesis, we predicted that dogs who show a high level of accuracy in copying the irrelevant action will also show good relationship quality to the human caregiver, from high numbers of gaze alternations as well as greeting, affiliative, synchronized and play behaviours in the relationship test.

## 2. Materials and Methods

### 2.1. Subjects

A total of 64 dogs completed both the overimitation and the relationship test. All dogs were family dogs, who were required to be at least a half year old, food motivated and naïve to the test-situation. Six dogs had to be excluded from the analysis due to demonstrator errors during the overimitation test. A list of the remaining 58 dogs and the copying accuracy levels of both actions (relevant and irrelevant) can be found in the Appendix A.

### 2.2. Experimental Setup

The overimitation tests were conducted at the Clever Dog Lab, Vienna. The testing room (6.0 × 3.3 m) was furnished with three cameras which were mounted at approximately 2 m of height and positioned to record the detailed performance of the dogs during the test. In addition, there was a chair for the experimenter to sit in during the demonstration (Figure 1).

For the test, a white wooden wall (150 × 100 cm) was installed to cover a corner in the room. This plate was modified with a cut-out (6 × 7 cm) covered by a white sliding door (10 × 9 cm) at a central position and 50 cm above the floor (Figure 2). The door could be moved via a brown wooden handle (4 cm long, 2 cm diameter) to the left or to the right (9 cm) and revealed a food receptacle. The receptacle could be filled with a treat. A white laminated poster (172 × 106 cm) was mounted at a distance of 130 cm from the sliding door but on the same wall. Two white, A4-sized sheets of paper (standard reprographic paper) with printed colour dots (9 cm in diagonal; one blue and one yellow) were glued to the poster 50 cm above the floor. To prevent an influence of scent cues from previously tested dogs, the poster and the white wooden wall, including the sliding door, were cleaned after every session with alcohol. The A4 sheets with the colour dots were replaced by freshly printed sheets for every dog. In addition, each caregiver was asked to touch a plain white paper prior to the test session, which was then glued to the door of the testing room.

The tests for the relationship between dog and partner took place on the same day as the overimitation test, after a 5-min break, in an outdoor area (25 × 13 m) on the campus of the Vetmeduni Vienna, with trees and a ground covered in grass (Figure 3). The area was surrounded by a chain-link fence. Subjects were undisturbed by passers-by as they were far enough away and the surroundings were quiet. All outdoor tests were recorded with a camera on a tripod.

### 2.3. Procedures

The overimitation test was conducted in the exact same way as in Huber et al. [19,21]. The caregivers of the test subjects had been instructed verbally and with a video example how to perform the demonstration, but were not informed about the aim of the study and our predictions. The paper which the caregiver was asked to touch before, and that was glued to the door of the testing room, served as a scent control to check whether the dogs would simply follow scent cues left by the caregiver.

At the beginning of an overimitation test trial, the experimenter brought the dog to the observer position. The dog was on a short leash and the experimenter was sitting behind it on a chair, with the human caregiver standing to the left. As soon as the dog sat calmly, the caregiver fed the dog a treat to gain its attention to then start the demonstration. The demonstrator attempted to show the two actions in a dog-like manner by using the same body parts that the dog would use. Therefore, the caregiver got down on hands and knees to touch the dots and open the sliding door with his/her nose (Figure 2). In contrast to Huber et al. [19], we did not divide the subjects into four different groups and only demonstrated the classical overimitation sequence in which the irrelevant action is followed by the relevant action [20,24]. The causally irrelevant action included touching the two coloured dots, first the blue then the yellow, with the nose. Next, the caregiver went to the white, wooden wall and demonstrated the relevant action. The demonstrator got down on hands and knees again and pushed the handle of the sliding door with their nose, to the left. The treat was taken out with the hand, shown to the dog and then, after turning around again, was placed back into the food box (covered by the demonstrator’s body and out of the dog’s view). Immediately after the demonstrator returned to the starting position, the dog was released by the experimenter. The dog was allowed to explore the room freely for two minutes and possibly copy the demonstrated actions.

The relationship tests have been conducted in the same way as by Cimarelli et al. [22,23], except for the social threat test, which was excluded as it was not used to determine the relationship by Cimarelli et al. [22]. The test consisted of four sub-tests (see Figure 3).

*Exploration of an unfamiliar outdoor area (3 min)*: When the dog and partner entered the area, the dog was released from the leash while the experimenter closed the gate. The situation aimed to simulate a visit at the dog park. The partner was instructed beforehand to behave as naturally as possible and to walk around in the area without calling the dog or giving commands. Responding to attention-seeking by the dog was allowed (looking at or talking to the dog).

*Separation (3 min)*: After the exploration phase, the dog was taken to a fenced corner inside of the area by the partner. To allow visual separation, the partner was then hiding behind a blanket at the opposite side of the enclosure. The separation phase started when the partner was behind the blanket and was out of the dog’s view.

*Reunion (3 min)*: The separation phase ended when the experimenter opened the fenced corner and allowed reunion with the partner, who was standing up and moving away from the blanket. The partner was allowed to respond to the greeting of the dog as usual, but was instructed not to call the dog. Additionally, he/she should not re-initiate greeting if the dog stopped the interaction, but rather behave as during the exploration. After 3 min, the dog was leashed and taken out of the area and behind a visual barrier.

*Novel object (3 min)*: While the subject was out of the area, the experimenter placed a novel object inside the enclosure. The object (either a plastic cube, a children’s toy tent shaped like a castle, a stuffed hippo or a big bag, all purchased from IKEA) was dangling from a rope. The rope was thrown over the branch of a tree and held by the experimenter on the other side, so it could be moved from the outside and out of sight of the dog. As soon as the experimenter was ready, the dog and its partner entered the enclosure again. The dog was released from the leash to start the novel object phase and the partner was instructed to behave in the same way as during exploration. After 3 min, the partner leashed the dog again and they left the area.

### 2.4. Data Analysis

Following Huber et al. [21], we categorized dogs’ behaviour during the overimitation test according to the copying accuracy level. In the case of the irrelevant action, we defined three levels of copying accuracy: (0) no touching of the dots, (1) touching of one dot and (2) touching both dots. In contrast to Huber et al. [21], we combined dogs that touched both dots in whatever order, as only one dog touched both dots in the wrong order and only two touched them in the correct way. In the case of the relevant action, we defined the following four levels of copying accuracy: (0) no touching of the apparatus, (1) touching of the apparatus, (2) opening the sliding door in the wrong way and (3) opening the sliding door in the correct way.

For the relationship test, we analysed whether the behavioural variables linked to reference and affiliation coded during the test would be predictive of dogs’ copying accuracy during the overimitation test by running two ordinal multinomial regression models. The variables “gaze towards the owner”, “gaze alternation between the object and the owner”, “affiliative behaviours”, “greeting” and “synchronization” (video coded following the definitions in [22,23]) were included as predictors, while “copying accuracy of the irrelevant action” and “copying accuracy of the relevant action” were included as response variables in the two models, respectively. We fitted models in R (version 4.0.2 [25]) using the *plor* function in the package MASS [26]. We compared the full models to null models, including solely the intercept with a likelihood ratio test [27]. Moreover, we calculated model stability by removing individual cases one at a time and comparing model estimates obtained from each subset to those of the full dataset. The models’ stability result was acceptable. Nonparametric bootstrapping (N = 1000 bootstraps) of the models’ coefficients and fitted values was used to obtain confidence intervals for each predictor.

## 3. Results

Of the 58 dogs who participated in the present study, three showed the highest level of copying accuracy of the irrelevant action, and 12 dogs touched at least one dot (Table 1). Twenty-one dogs managed to open the sliding door of the food compartment (nine did so in the same way as the owner demonstrated), while twenty only touched it.

Overall, it seems that, on average, the three dogs showing the highest level of copying accuracy of the irrelevant action also showed longer gaze towards the owner, more frequent gaze alternation between the owner and the object during the novel object test, more affiliative behaviours, longer greeting, and longer synchronization in the relationship test than dogs showing a lower level of copying accuracy (Table 2, Figure 4). However, the full–null model comparison for the copying accuracy of the irrelevant action result was non-significant (χ^2^ = 1.28, df = 5, *p* = 0.93), likely due to the low number of dogs showing the highest level of copying accuracy.

In contrast, the behaviours shown during the relationship test did not seem to vary linearly with the copying accuracy of the relevant action (Figure 5). Additionally, in this case, the null–full model comparison result was non-significant (χ^2^ = 5.32, df = 5, *p* = 0.38).

## 4. Discussion

This study aimed at testing the hypothesis that dogs would copy irrelevant, causally unnecessary actions of a human demonstrator, especially if they have a close and positive relationship with her/him. Earlier studies revealed a significantly higher number of dogs doing so when the human demonstrator was the caregiver of the dog than when the demonstrator was unfamiliar to the dog [19,21]. Likewise, in the study by Johnston and colleagues [20], a high number of dogs copied the irrelevant actions from a human experimenter who had repeatedly rewarded the subjects with food before the test. Here, we further explored this possible link by correlating informative and affiliative components of the human–dog relationship with the probability and accuracy of overimitation. We found no significant positive correlation. The absence of a significant effect was probably due to the surprisingly low number of dogs showing high-accuracy overimitation (see Appendix A): from the 58 dogs tested, only 3 showed the highest level of copying accuracy of the irrelevant action (Level 2), and 12 dogs touched at least one dot (Level 1).

Still, the few dogs that showed the highest level of copying accuracy of the irrelevant action were, on average, the ones with the highest values in all five behaviours relating to reference and affiliation. In the relationship test, they overall gazed longer at the caregiver, alternated more frequently between caregiver and novel object, and even showed more affiliative behaviours as well as longer greeting and synchronization, all in comparison to the dogs that did not copy the irrelevant action or showed a lower level of copying accuracy. Interestingly, nothing similar emerges if we look at the potential relationship between these owner-directed behaviours during the relationship test with the copying accuracy of the relevant action. Therefore, we cannot conclude that dogs with higher owner-directed and affiliative behaviours are in general more inclined to copy her/his actions, but that maybe dogs do so in the case of irrelevant actions (although this did not emerge statistically from our data, likely because of the low number of dogs showing the highest copying accuracy).

Reasons for this reduction in copying behaviour compared to the results of former studies, which were precisely replicated here, will be discussed in the following.

One factor that might have played an inhibiting role on copying motivation was the high ambient temperature at the time of testing. The data for Huber et al. [19] had been collected during autumn, whereas this study tested the dogs during summer. As there is no air conditioning in the Clever Dog Lab, temperatures were generally very high, which could have negatively affected dogs’ overall motivation to move or their ability to concentrate.

Another difference between the two studies was the short attention test we administered shortly before the overimitation test in the original study, but not here. Arguably, dogs might become prepared to watch carefully if they have exhibited six object-permanence trials shortly before the observation experiment [19]. After such careful observation, they would then be more likely to show high accuracy copying than when being less attentive. For both irrelevant and relevant actions, we found most dogs showing low and moderate levels of copying accuracy, i.e., approaching but not touching (the dots or the apparatus) or opening the sliding door. Even in the case of the relevant action, we found only about a third of the dogs copying the target behaviour. Although 31 dogs approached the sliding door, implying a certain amount of interest, only 21 completed the action by pushing the door to the side.

Taken together, these two differences (i.e., the hot temperature and the absence of an attention test beforehand) to the previous, generally very similar study [19], might have made dogs selective and focus only on what was relevant. It might have caused the dogs to be more willing to search for the treat than spend unnecessary energy overimitating in the hot conditions of this study. Hence, they copied what was relevant to get the treat but paid less attention to what was irrelevant for it. The social motivational aspects, which we prefer to attribute to the willingness to overimitate, might have been pushed down by these less favourable conditions. There is also the possibility that previous studies showing a higher number of dogs overimitating [19] overrepresented the number of dogs able to overimitate in the general dog population. Of course, all of this is very speculative, and only further testing can clarify these issues.

Of course, we cannot dismiss the possibility that the readiness or willingness of pet dogs to copy non-functional actions of the caregiver are not driven by affiliative reasons, i.e., to promote affiliation with the demonstrator by behaving as she/he likes to behave, but are driven by normative reasons, by behaving the way the demonstrator wants the dog to behave [18]. Especially if the demonstrator remains present during the test, the dog may feel a kind of normative pressure to behave in the same way as demonstrated [28,29,30]. It is possible that dogs interpret the situation as a kind of training in which they are taught to behave in the demonstrated manner. Pet dogs receive many different kinds of training in the home environment, and many have also received specific trainings in dog schools. It has been assumed that the way in which a dog was trained in the past may also affect its future aptitude and motivation to learn [31] and therefore its performance at novel training tasks, as well as its general obedience. Dogs are not only trained with verbal commands, but also with gestures and by showing the target movements. In the case of the dog’s understanding of human pointing gestures, researchers discussed the possibility that following communicative pointing in a discriminative manner results from the perception of the pointing gesture as an imperative signal, ordering them where to go [32,33]. For instance, in a study by Szetei et al. [34], the majority (79%) of dogs followed the human gesture to an empty cup and ignored their own, accurate information. Especially if the gesture is accompanied by ostensive-communicative cues, such as making eye-contact and saying the dog’s name, the dog may perceive the situation as a teaching or training setting. Perhaps our overimitation test, in which the caregiver initiated by feeding the dog a treat to gain attention, has also been interpreted by the dog as training, and therefore they followed the caregiver to the same locations, independently from the relationship the dog had with the caregiver.

## 5. Conclusions

In conclusion, our study cannot support the hypothesis of a positive correlation between overimitation and relationship quality between dog observers and the caregiver demonstrators, perhaps due to the low number of overimitators among the sample of this study in comparison to the previous study [19]. Further studies will need to be conducted to explain how dogs interpret the demonstration of perceivably causally unnecessary actions in relation to the goal of an action-sequence performed by a human model. Undoubtedly, overimitation research in dogs is still in its infancy.

## Figures and Tables

**Figure 1 animals-12-00326-f001:**
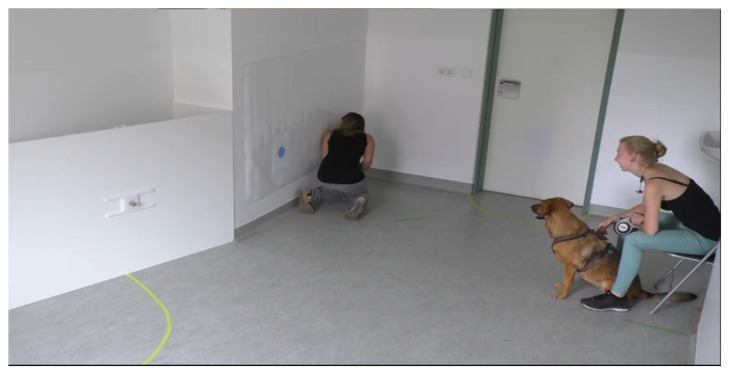
Snapshot of an overimitation testing situation, showing a dog subject sitting next to the experimenter and watching the demonstration by its caregiver.

**Figure 2 animals-12-00326-f002:**
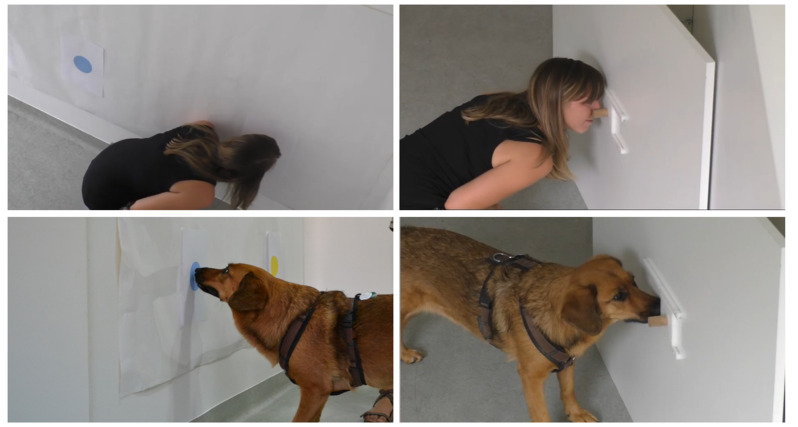
Snapshots of a caregiver (**above**) demonstrating the irrelevant (**left**) and relevant action (**right**), and of the caregiver’s dog (**below**) when copying these actions.

**Figure 3 animals-12-00326-f003:**
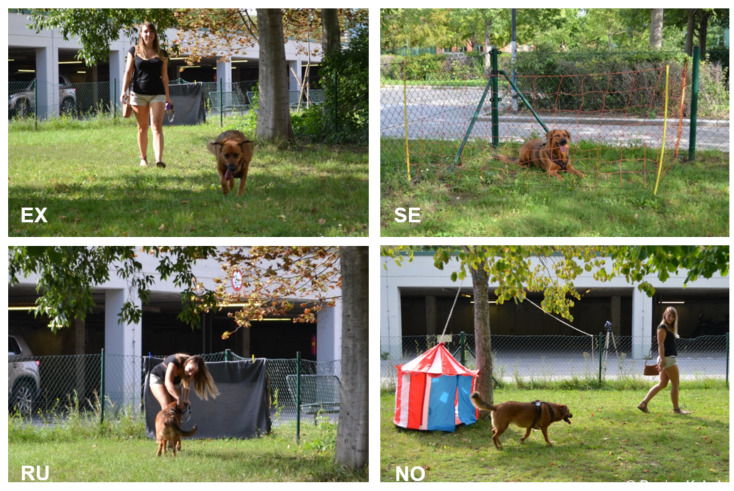
Snapshots of the four phases of the relationship test. Exploration (**EX**), Separation (**SE**), Reunion (**RU**) and Novel object test (**NO**).

**Figure 4 animals-12-00326-f004:**
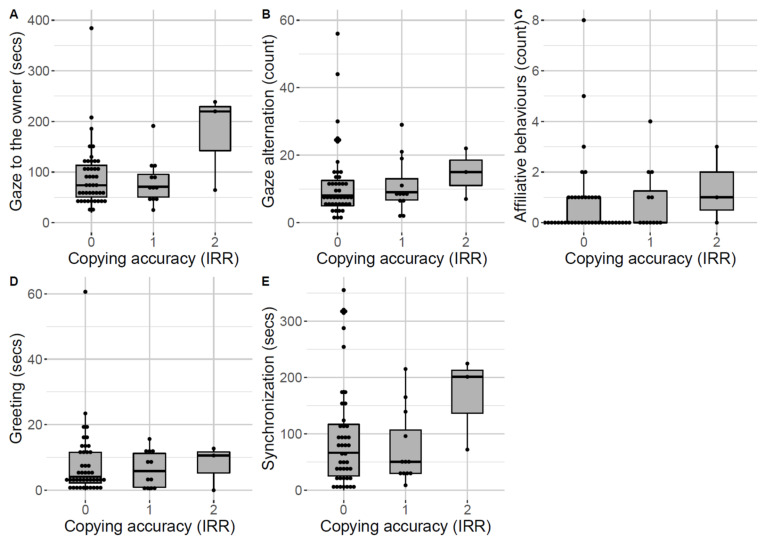
Owner-directed behaviours during the relationship test across dogs showing the different levels of copying accuracy of the irrelevant action (IRR) during the overimitation test: (**A**) Gaze to the owner; (**B**) Gaze alternation; (**C**) Affiliative behaviours; (**D**) Greeting; (**E**) Synchronization. Median and interquartile range (IQR; represented by the box), 25th percentile + 1.5 IQR, and 75th − 1.5 IQR (represented by the lower and the upper whiskers, respectively). Dots represent outliers.

**Figure 5 animals-12-00326-f005:**
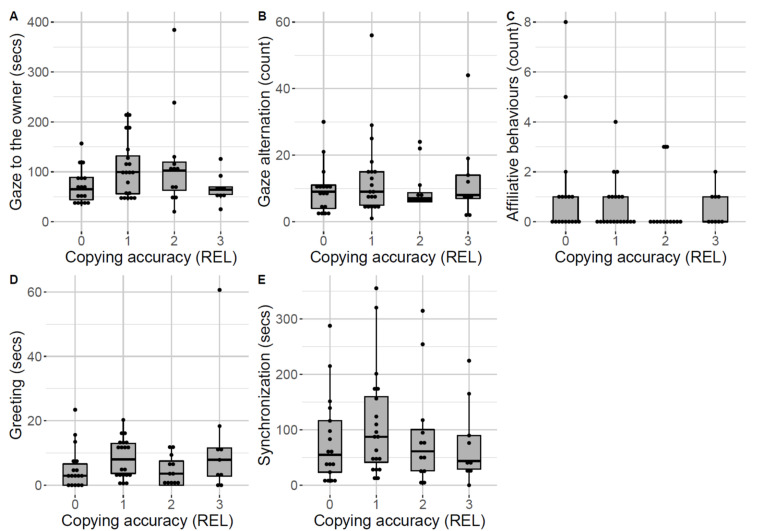
Owner-directed behaviours during the relationship test across dogs showing the different levels of copying accuracy of the relevant action (REL) during the overimitation test: (**A**) Gaze to the owner; (**B**) Gaze alternation; (**C**) Affiliative behaviours; (**D**) Greeting; (**E**) Synchronization. Median and interquartile range (IQR; represented by the box), 25th percentile + 1.5 IQR, and 75th − 1.5 IQR (represented by the lower and the upper whiskers, respectively). Dots represent outliers.

**Table 1 animals-12-00326-t001:** Levels of copying accuracy: Absolute and relative numbers of dogs that copied the two actions at different levels of accuracy; M male; F female.

**Level**	**Irrelevant Action**	**M**	**F**	**Sum**	**%**
0	no touching of the dots	24	19	43	74.1
1	touching one dot	8	4	12	20.7
2	touching both dots	0	3	3	5.2
	**Relevant action**	**M**	**F**	**Sum**	
0	no touching of the apparatus	2	4	6	10.3
1	touching of the apparatus	18	13	31	53.4
2	opening the sliding door in the wrong way	9	3	12	20.7
3	opening the sliding door in the correct way	3	6	9	15.5

**Table 2 animals-12-00326-t002:** Descriptive statistics for each owner-directed behaviour during the relationship test across dogs showing the different levels of copying accuracy.

Action	Level of Copying Accuracy	Owner-Directed Variable	Median	SE
Irrelevant (IRR)	2	Gaze to the owner	219.84	55.19
	1		70.74	12.72
	0		73.84	9.48
	2	Gaze alternation	15.00	4.33
	1		9.00	2.32
	0		8.00	1.63
	2	Affiliative behaviours	1.00	0.88
	1		0.00	0.37
	0		0.00	0.00
	2	Greeting	10.60	3.93
	1		5.88	1.66
	0		4.12	1.58
	2	Synchronization	200.96	47.40
	1		50.32	18.79
	0		66.36	14.07
Relevant (REL)	3	Gaze to the owner	64.40	9.48
	2		102.26	28.88
	1		99.48	12.74
	0		65.04	8.41
	3	Gaze alternation	8.00	4.29
	2		7.00	1.83
	1		9.00	2.76
	0		9.00	1.76
	3	Affiliative behaviours	0.00	0.24
	2		0.00	0.34
	1		0.00	0.23
	0		0.00	0.52
	3	Greeting	7.88	6.30
	2		3.58	1.33
	1		8.00	1.38
	0		2.96	1.59
	3	Synchronization	43.72	25.57
	2		61.06	28.16
	1		87.36	21.56
	0		54.92	19.35

## Data Availability

The data supporting reported results can be found in the Appendix A.

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
