# Peer review of "Overimitation in Dogs: Is There a Link to the Quality of the Relationship with the Caregiver?"

_animals, 2022, doi:10.3390/ani12030326_

Round 1

Reviewer 1 Report

Comments to Authors:

This is a relevant and well written paper investigating whether the relationship quality between dog and caregiver would be linked to the tendency of overimitating an irrelevant action. The study presents an important contribution since it provides knowledge about how dogs learn and how the affectional bond facilitates the learning. The experiment was carefully designed; however, some doubts were raised about the methods that should be addressed. I believe that after clarifications and a minor review, the manuscript will be suitable for publication in Animals.

General comment

It was not clear what is the reason to have the relevant action in this study since the focus was the overimitation (of an irrelevant action). At first it seemed that relevant action would be used as a kind of control condition, however there was no comparisons between results obtained in irrelevant and relevant conditions. They were analyzed separately. The reason to include the relevant action in the study could be better clarified.

Simple summary and Abstract

This is the first contact of the reader with the study; even saying that the overall results were not significant, when it is stated that data revealed that dogs who overimitated seemed to show more referential and affiliative behaviours towards the owner, there is a risk that the reader will understand that these specific associations have actually been found, then I would recommend to say that this tendency was observed based on three dogs who overimitated.

Introduction

In the last paragraph of the introduction, letters A, B and E were used to describe the actions in the overimitating test:

“In the overimitation test, we confronted the dogs with the demonstration of A; a causally superfluous, functionally opaque or irrelevant action, B; an effect-relevant or causally relevant action, and E; the effect or goal.”

The reader will expect that these letters would be used ahead in the manuscript, but since they were not used, I suggest removing from here and simply call them as irrelevant action and relevant action.

Methods / Discussion

Why the study stablished the order: overimitation test followed by relationship test? It is important to explain this choice and discuss whether this order could influence results or not.

Regarding the overimitation test specifically, why the procedure stablished the order: irrelevant action followed by the relevant action? It is also important to explain this choice and whether this order could influence results or not.

Results

Table 1 show the absolute and relative numbers of dogs that copied the two actions at different levels of accuracy separated by sex of the dog. Was the variable sex of the dog included in the full model?

Author Response

The reviewer is right that there is a risk that the reader will understand that these specific associations have actually been found. And we followed the recommendation for saying that this tendency was observed based on three dogs who overimitated at the highest level of accuracy.

We now wrote at the end of the abstract: “Although we found no significant correlation between the two test results, our data might suggest that, on average, the three dogs who overimitated seemed to show more referential and affiliative behaviours towards the owner than dogs who showed less or no copying of the irrelevant action. Notably, as a group, those dogs that showed the highest level of copying accuracy of the irrelevant action showed the highest level of gazing and synchronization towards the owner.”

Following the suggestion we removed at the end of the introduction the letters A, B and E and simply mentioned the irrelevant action and relevant action.

We now mentioned the order of the two tests (first the overimitation test, then the relationship test) that have been conducted on the same day. The reason for it was just a practical one: we wanted them to be “fresher” for the overimitation test, to avoid that they would not pay attention during the demonstration.

Concerning the order of actions in the overimitation test, we now added that the order of these two actions followed the classical procedure (e.g., Horner & Whiten 2005; Johnston et al. 2017). The relevant action is the one that allows to reach the goal, it is logically necessary that the irrelevant action happens before that. Furthermore, we did not find significant differences if the order were reversed, as was done in our previous studies (Huber et al., 2018, 2020).

Concerning the question if the variable sex of the dog was included in the full model: no, it was not, we had no specific hypotheses about it, but we thought to include the data in the Table 1 because it does not harm to have the data presented in the table, but it could be useful for a potential future meta-analysis and some readers may be interested to see those data.

Reviewer 2 Report

Overview

I was glad to be asked to review this paper on the relationship between overimitation in dogs related to the quality of the dog’s relationship with its caregiver. Dogs were selected to take part in 2 batteries of tests, with the behaviors being videorecorded for analysis later. While there was no significant findings, there were some tendencies toward a relationship between a higher affiliative relationship and the dog performing imitative behaviors.

General comments

This paper was very well-written, with a clear hypothesis and methodology, along with results and a complete discussion. It can benefit from displaying some of the results in table form, due to the lengthiness in the statistical analyses in lines 250-280.

Introduction

Your introduction was very complete and appeared to evaluate all of the available literature on this subject. One option to explore is the owner’s attachment to the pet, via the Lexington Attachment to Pets Score (LAPS).

Methodology

Your methodology was well-thought-out and clearly presented. The pictures provided a solid supplement to the presented information, making it very clear. While not a statistician, I can appreciate that you modeled the behaviors and made sure that your modeling was stable for this paper.

Results

I appreciate that you presented the results as you did, not overstretching the results to fit your hypothesis. I like the graphs on page 7. I believe that your results would be clearer to interpret if the data were to be presented in table format.

Discussion

Your discussion brought forth the drawbacks for your study in a measured manner, without stretching the results to fit your hypothesis. One suggestion would be to identify whether the selection of these dyads were different than at other times. Additionally I would suggest at least addressing the quality of the attachment to pets via the LAPS. I do understand your investigation of the dog’s relationship, but it’s worth mentioning the owner’s relationship.

References

Your review of the pertinent literature was well done and comprehensive.

Author Response

We agree with the reviewer that it can benefit from displaying some of the results in table form. In Table 1 we have already the data from the overimitation test, therefore we added the (new) Table 2, providing descriptive statistics for each owner-directed behaviour during the Relationship test across dogs showing the different levels of copying accuracy.

Thank you for the suggestion of exploring the owner’s attachment to the pet via the Lexington Attachment to Pets Score (LAPS).  But the aim of the study was to focus on the relationship from the perspective of the dog, not the owner, as the connection with overimitation would have been very indirect: owner attachment -> owner behaviour -> relationship from dog’s perspective -> overimitation. Still, we will consider the relationship from the perspective of the human in future studies.

Concerning the selection of dogs and the mentioning the owner’s relationship. Subjects’ selection was not different from previous studies: dog owners were invited to participate in the study either thanks to contact details provided spontaneously by owners to the Clever Dog Lab or thanks to recruitment through social media. The sex, age and breed distribution were comparable to the one of previous studies carried out in our lab. And, again, future studies will have to look at the dog-human relationship from the perspective of the human as well.

Reviewer 3 Report

In my opinion, this is an interesting study, reported in a well-written manuscript. I only have a couple of minor issues with clarity and some discussion points I’d like to see addressed.

Simple summary/Abstract: Works well, fine overview, faithful to findings, and easy to grasp.

Minor issues:

  • Very similar, are both required?
  • “Not significant” [but] “revealed” [but] “seemed to” rings a bit off. Could it be phrased better?

Introduction: Fine. Sufficient background on the problem of over-imitation in dogs, previous studies in the area, and how these suggest the present hypothesis about owner-affiliation as possible factor.

I have no issues with this section.

Method: A well designed experiment and generally a transparent exposition with good use of illustrations. I’m not familiar with the boot strapping method used so I cannot comment on that.

Minor issue:

  • Why four novel objects (line 202)? How was the choice between them made, and did it have possible implication for results? Please explain.

Results: Thoroughly reported. Table 1 is relevant, and the box plots (Figure 4 and 5) work well for conveying the main findings.

Minor issues:

  • The sections spelling out the numbers (lines 251-275) are hard to follow. I suggest putting the numbers in a table (or two tables) instead, and let the text summarize only.
  • The shift in the middle of a paragraph (line 263) is somewhat confusing. It goes from commenting on irrelevant action (reported in the previous section) to reporting about relevant action (immediately following lines), which took me several re-readings to catch. I suggest adding a section division in the middle of line 263. Also, I’m not sure that “instead” is the right transition word.

Discussion: Generally, a clear summary and interpretation of the findings. The lower-than-expected level of over-imitation in the present sample of dogs produced vague results, and this was - relevantly - discussed at length. Much of this rests on comparison to the authors’ previous study, which found a higher over-imitation frequency. The discussion mentions a couple of differences between the two studies that may partly explain the difference. I would like to see a couple of other issues raised as well.

Minor issues with clarity of the comparison between two studies, section 316-326:

  • Lines 320-322 are hard to follow. Please rephase.
  • In line 325-326, do numbers refer to the present or the previous study? Could this be clearer? I think it may be helpful to directly comparing results from the two studies, using percentages.

Discussion points I would like to see:

  • Only few studies have demonstrated over-imitation in dogs. I would like to see the discussion explicitly address – if only briefly - whether previous studies found “too much” rather than this one finding “too little”. Is over-imitation in dogs certainly a thing or might it be a methodological artefact?
  • Was the owner’s usual training method asked about and controlled for? Should it be in future studies? I would expect the dogs' training experiences to impact their reaction to the experiment, e.g., whether they were used to punishment versus free shaping versus do-as-I-do...
  • Might another test of affiliation work better for the type of affiliation that is relevant here?

Author Response

We rephrased the last part of the abstract.

Concerning the question of why we used four different objects in the Novel Object Test, we did so following the advice of our expert statistician. To increase the generalizability of the results it is generally recommended to use more than one stimulus or object. 

In the Results we deleted the detailed list of median data and put the numbers in a new table (Table 2). And they can be read from the figures as well.

We exchanged “Instead” against “In contrast”, and shifted the previous sentence to the paragraph before, adding the section division after the sentence about the contrast, as being suggested by the reviewer.

We rephrased the sentence that was previously lines 320-322. And yes, the numbers in the last sentence of this paragraph refer to the present study.

Concerning the differences to our previous studies, we discussed two possibilities already, the hot temperature and the absence of an attention test beforehand. Generally, it is difficult to conclude whether the first study (Huber et al. 2018) found too much of overimitation, but we see no reason why to doubt the previous results. But we added now a sentence “There is also the possibility that previous studies showing a higher number of dogs overimitating [19] overrepresented the number of dogs able to overimitate in the general dog population.”

Concerning the question whether the dogs' training experiences impact their reaction to the experiment, e.g., whether they were used to punishment versus free shaping versus do-as-I-do: The influence of the dogs’ training history (such as target training at home or touch-screen or eye-tracking training in the Clever Dog Lab) has been examined and discussed in our first study (Huber et al. 2018), but due to the low number of individuals, the statistical test did not reach significance. In the present study we did not investigate this issue, due to the even lower number of individuals who overimitated and the fact that the focus of this study was another one (relationship).

Concerning the question whether another test of affiliation might work better for the type of affiliation that is relevant here we would say yes, this might be, and actually we are planning such studies at the moment.